# Genotypic Characteristics and Antimicrobial Resistance of *Escherichia coli* ST141 Clonal Group

**DOI:** 10.3390/antibiotics12020382

**Published:** 2023-02-13

**Authors:** Audrey Emery, Didier Hocquet, Richard Bonnet, Xavier Bertrand

**Affiliations:** 1Hygiène Hospitalière, CHU, 25030 Besançon, France; 2Bactériologie, CHU Nice, 06202 Nice, France; 3Chrono-Environnement, UMR 6249, CNRS Université de Franche-Comté, 25000 Besançon, France; 4UMR 1071, Université d’Auvergne, 63000 Clermont-Ferrand, France; 5Bactériologie, CHU Clermont-Ferrand, 63000 Clermont-Ferrand, France

**Keywords:** *Escherichia coli*, ESBL, epidemic lineage, WGS, O2:H6, ST141

## Abstract

*Escherichia coli* ST141 is one of the ExPEC lineages whose incidence is rising in France, even if no epidemic situation involving multidrug resistant isolates has been reported so far. Nonetheless, in a 2015–2017 monocentric study conducted in our French University hospital, ST141 was the most frequent lineage after ST131 in our collection of phylogroup B2 ESBL-producing *E. coli*. The genomes of 187 isolates representing ST141 group, including 170 genomes from public databases and 17 from our local collection, of which 13 produced ESBL, were analyzed to infer the maximum likelihood phylogeny SNP-based (Single Nucleotide Polymorphism) free-recombinant tree defining the ST141 population structure. Genomes were screened for genes encoding virulence factors (VFs) and antimicrobial resistance (AMR). We also evaluated the distribution of isolates according to their origin (host, disease, country) and the distribution of VFs or AMR genes. Finally, the phylogenic tree revealed that ST141 isolates clustered into two main sublineages, with low genetic diversity. Contrasting with a highly virulent profile, as many isolates accumulated VFs, the prevalence of AMR was limited, with no evidence of multidrug resistant emerging lineage. However, our results suggest that surveillance of this clonal group, which has the potential to spread widely in the community, would be essential.

## 1. Introduction

*Escherichia coli* is the predominant aerobic bacterium in the normal gut microbiota of humans and vertebrates as well as a major human pathogen [1]. Indeed, *E. coli* strains can cause both extra-intestinal pathologies (urinary tract infection, intra-abdominal or pulmonary infection as well as newborn meningitis or bacteraemia) and intestinal infections [2]. Currently, unlike these latter one (Intestinal Pathogenic *E. coli*, InPEC), for which virulence factors have been clearly identified and for which pathovar classification seems easy, the classification of ExPEC (Extraintestinal Pathogenic *E. coli*) strains is still subject to discussion, as no disease-specific virulence genes have been identified. It is the combination of different virulence factors, which may be numerous in these strains, that may explain their pathogenicity. Indeed, most, if not all, strains that may cause extra-intestinal infections have different genes encoding for virulence factors, such as adhesins, toxins, protectins and iron capture systems. However, the conditions under which *E. coli* strains emerge from their intestinal reservoir to cause extraintestinal infections, remain largely unknown as well as the reasons for the success of some pandemic lineages. Indeed, phylogenomic approaches have demonstrated that only four sequence types or sequencing type complexes (STcs), responsible of extraintestinal infections (STc131, STc73 and STc95 belonging to phylogroup B2, and STc69 belonging to phylogroup D) were always observed in epidemiological studies and were thus named “the big four ExPEC clones”. Although not part of the big four ExPEC clones, *E. coli* ST141 is regularly reported as one of the most represented ExPEC [3]. Furthermore, among ExPEC, *E. coli* ST141 is singular: it has recently been hypothesized that the ST141 *E. coli* lineage genome would be very susceptible to recombinations, making it capable of acquiring and expressing specific genes of InPEC thus conferring a heteropathogen status [4]. The appearance of such hybrid clones could reshuffle the deck within the pathotype classification widely used to describe *E. coli* populations. Another matter of concern is the ability of E. *coli* to gain antibiotic resistance determinants given that *E. coli* may easily spread, as occurred with the pandemic of highly drug-resistant *E. coli* sequence type 131 (ST131) H30 sublineage [5]. To date, no epidemic situation involving multidrug resistant ST141 isolates has been reported [6]. Nonetheless, in a 2015–2017 monocentric study conducted in our French University hospital, consisting of collecting all non-duplicate extended-spectrum beta-lactamase producing *E. coli* (ESBLEc), we identified that ST141 was the second most frequent lineage in our collection of phylogroup B2 ESBLEc, accounting for 8.7% of ESBL-producing isolates, while ST131 lineage, the most frequent lineage, accounted for 55.3%, and isolates belonging to other STs each accounted for less than 3% of isolates [7].

Here, we used a comprehensive data set of ST141 genomes available in databases to characterize population structure of this clonal group with the aim of identifying the emergence of a multidrug resistant sublineage.

## 2. Results

### 2.1. ST141 Collection (Appendix A)

Among the 187 isolates analyzed, the geographic origin was known for 154 (82%): 108 isolates originated from European continent (mainly from France, United Kingdom, and Germany), 40 from American continent (North and South) and 6 from Asian continent. Most of the isolates (72%, 135/187) have been collected from humans, while non-human strains (i.e., environment, animal, and food) accounted for 15% (28/187) of the collection.Available genomes came from strains isolated between 1988 and 2019 and the sample date was known for 139 of which. A majority of strains were collected after 2010’s (71.2%, 99/139).Among the 135 human-isolated strains, information about source sample was known for 100 isolates: 36% (36/100) came from urine, 33% (33/100) from blood culture, 23% (23/100) from feces (of which 13/23 were responsible of diarrhea) and 8% (8/100) from various clinical samples (of which three came from neonatal infections).

### 2.2. Whole Genome-Based Typing and SNP-Based Recombination-Free Phylogenetic Tree of ST141 E. coli Strains 

In silico MLST (Multi-Locus Sequence Typing) and phylogrouping confirmed that all selected isolates belonged to ST141 and B2 phylogroup. Within this ST141 clone, we wanted to assess the diversity of serogroup O and the fimH allele, which are two hotspots of recombinations, to be more discriminating [2]. Surprisingly, the ST141 clone has a low O-serogroupe diversity, as all strains tested belonged to O2:H6 serotype. The fimH allele typing, which is based on minor sequence variations, allowed to identify fimH5 (115/187, 61.5%), fimH14 (48/187, 25.7%) and fimH350 (10/187, 5.3%) as major fimH subtypes. Then, 16 minor fimH subtypes (i.e., shared by two or less isolates) were identified (Appendix A, Figure 1) [8].Sublineages were indicated by edge coloring. Color strip corresponded to source origin. For easy readability, branches with fewer leaf nodes have been displayed on the top. Only bootstraps with value > 80% of confidence were displayed (blue burred nodes). SNP-based analysis displayed two main sublineages (S1 and S2). Whereas S1 (green edges) appeared relatively homogeneous, with highly conserved *fim*H5 subtype (115/118), S2 (blue edges) *fim*H subtyping appeared less homogeneous, with predominant *fim*H14 (48/69) and *fim*H350 (10/69) subtypes. Isolates from local collection (colored in orange) are scattered throughout the phylogenetic tree. Most of them (15/17) belonged to S1 while only two strains (S89 and S96) belonged to S2.The SNP-based recombination-free phylogenetic tree depicted two main sublineages S1 and S2, clustering 118 and 69 isolates, respectively (Figure 1). S1 potentially appeared recently as a result of a *fim*H subtype conversion (Figure 1). Indeed, all isolates belonging to *fim*H5 clustered into S1 (115/118, 97.5%). The three remaining S1 isolates belonged to *fim*H7 (2/118, 1.7%) and *fim*H1497 (1/118, 0.8%) subtypes.Isolates from our local collection mostly belonged to S1 (15/17) and six of them exclusively grouped together. Isolates S86, S87, S80, S82, and S92 were indeed separated to each other by a maximum of 14 SNPs while S84 was distant with 26 SNPs. Finally, only two isolates, S89 and S96, did not belong to S1 (Appendix A, Figure 2).Isolates from urines were more represented in S1 than in S2 (S1: 38%—31/81 vs. S2: 11%—5/47, *p* < 0.01). Conversely, isolates responsible for diarrhea (S1: 4%, 3/81 vs. S2: 22%—10/47) preferentially belonged to S2 (*p* < 0.01) (Table 1).Interestingly, 33 isolates of collection and previously studied by Gati et al., we noted a good correlation between the L1/L2 Gati’s lineages and our S1/S2 distribution, as all isolates belonging to L2 clustered in the S1 sublineage and possessed the *fim*H5 subtype. Conversely, all isolates belonging to L1 clustered in the S2 sublineage, with different *fim*H subtypes (7 of subtype *fim*H14; three of subtype *fim*H350, two of subtype *fim*H76 and one of subtype *fim*H 674) [4] (Appendix A, Figure 1).

### 2.3. Antimicrobial Resistance Genes in E. coli ST141 (Appendix A, Figure 3)

Regarding resistance to β-lactams associated with extended-spectrum β-lactamase, a total of 35 strains (35/187, 18.7%) carried various bla_ESBL_ genes. Class A bla_CTX m_ genes were the most prevalent ESBL genes (80.0%, 28/35) with bla_CTX-M-14_ being the most frequent (7.0%, 13/187) followed by bla_CTX-M-15_ (3.7%, 7/187), which was most often associated with the broad spectrum β-lactamase bla_TEM-1_ gene (3/7, 43%), and bla_CTX-M-1_ 2.1% (4/187). Of note, 13 out of these 28 ESBL-producing isolates came from our local collection. No statistical difference relating to the ESBL distribution was observed between S1 (22.9%, 27/118), and S2 (11.2% 8/69, *p* = 0.08). We did not retrieve bla_ESBL_ genes in ST141 isolates recovered from the environment. Additionally, two isolates harbored carbapenemase-encoding genes bla_OXA-48_ (Ec81) and bla_VIM-1_ (C04) and one strain (Ec110) isolated from poultry had colistin resistance gene mcr-1 (Appendix A, Figure 2).

**Figure 3 antibiotics-12-00382-f003:**
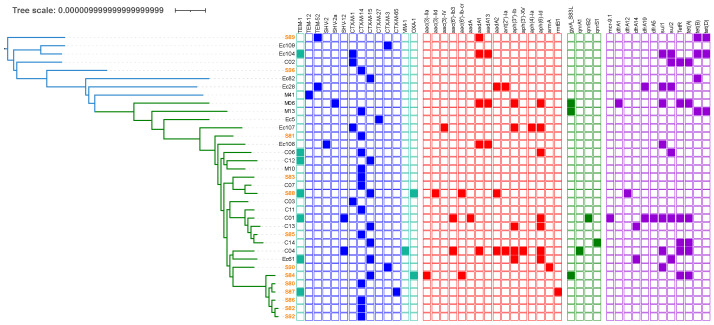
AMR genes distribution among *Escherichia coli* ST141 genomes with ESBL gene. Sublineages were indicated by edge coloring (S1, green edges, S2, blue edges). The presence/Absence of genes encoding AMR were indicated by colored squares (blue: β-lactam resistance—ESBL were shown in dark blue; red: aminoglycosides resistance; green: quinolone resistance; purple: miscellaneous).

Resistance genes to aminoglycosides were found in 21.9% (41/187) of the isolates and many of them (82.9%, 34/41) accumulated two or more resistance genes. The *aph*(6)-Id and *aph*(3″)-Ib, genes were the most common genes in our collection, as they were present in 15.0% (28/187) and 12.3% (23/187) of the isolates, respectively. We found no difference in the distribution of resistance genes to aminoglycosides between isolates of human and non-human origin (*p* = 0.6179) or between urine, blood or stool origin. Resistance genes to aminoglycosides were equally distributed between S1 (25.4%, 30/118) and S2 (15.9%, 11/69, *p* = 0.1462) (Appendix A). 

Resistance to quinolones was rare, as mutations in quinolone resistance-determining regions (QRDR) were found in nine isolates (4.8%, 9/187) and Plasmid-Mediated Quinolone-Resistance (PMQR) genes were present in only 2.1% (4/187) for qnr genes and 1.1% (2/187) for *aac*(6′)-Ib-cr gene. One isolate (S84) harbored both *aac*(6′)-Ib-cr and *gyr*A S83L mutation. Globally, the frequency of quinolone resistance (QRDR and PMQR) was not statistically different between S1 (10.2%, 12/118) and S2 (2.9%, 2/69), (*p* = 0.09) (Appendix A).

To note, all 13 ESBL *E. coli* strains in our collection were tested in vitro (diffusion susceptibility testing according to EUCAST/CA-SFM recommendations in force at the time of the establishment of the strain collection) with a perfect correlation with results obtained in silico.

### 2.4. Virulence Factors in E. coli ST141 (Table 2 and Appendix A)

Overall, we did not find a strong association between the distribution of VFs and the geographical origin, the origin (human or non-human), and the type of clinical sample. However, some differences in the distribution of VFs in S1 and S2 were statistically significant and displayed in Table 2. Thus, *stx*2 gene was more frequent in the S2 sublineage (due to the predominance of strains causing diarrhea in this sublineage (S1 = 5/118; S2 = 12/118, *p* = 0.004)), while the Enteroaggregative *E. coli* (EAEC) *ast*A gene previously described to exclusively belong to strains of the L2 (i.e., S1 in this study) sublineage was significantly predominant in S1 sublineage [4]. However, the *ast*A gene was also recovered in two isolates (Ec116 and C08) from our collection, classified as lineage S2, as they belonged to *fim*H14 subtype. Of note, while this gene is supposed to be typical of EAEC (Enteroadhesive *E. coli*, responsible of diarrhea), we showed that *ast*A gene was statistically more present in isolates from urine (23/36, 63.9%) than in isolates from stool (5/18, 21.7%, *p* = 0.0028). Finally, the invasion gene *ibe*ABC was exclusively present in S2 sublineage.Assuming that the association of *hek*, *pap*, *cnf*1 and *hly*A genes could lead to the hypothesis of the presence of PAI II_536-like_, we sought to highlight the strains in our collection that possessed these four genes. Finally, a total of 102 isolates (S1 = 69/118, 58.5% and S2 = 33/69, 47.8%, *p* = 0.17), from various origins (blood = 15/33, 45.5%, urine = 28/36, 77.8%, stool = 17/23, 73.9%, other or unknown = 42/95) could possess the PAI II_536-like_ UPEC virulence factor [4]. Statistically, PAI II_536-like_ was less frequent in strains isolated in blood than those isolated from urine (*p* = 0.007). No such difference could be demonstrated between stool and urinary strains (statistically insignificant, *p* > 0.01) (Appendix A).Based on the evolutionary model of STEC/UPEC hybrid proposed by Gati et al., we then searched for an association of the gene *stx*2 with other specific genes (PAI II_536-like_ or EHEC-*hly* gene) [4]. Among the four additional isolates owning *stx*2 and not previously described by Gati et al. (Ec22, Ec23, Ec93 and Ec166), only one of them had both *stx*2 and PAI II_536-like_ (i.e., Ec22), suggesting that the three others might have lost PAI II_536-like_ (Appendix A). However, in our study collection, we have not identified any strain with EHEC-*hly* gene, except for the strain previously described in Gati’s study (N011) [4].

**Table 2 antibiotics-12-00382-t002:** Virulence genes and lineage distribution of *E. coli* ST141 isolates.

Pathovar	Virulence Genes	S1 ^1^ (%)	S2 ^1^ (%)	*p*-Value
ExPEC ^1^	*tia*/*hek*	80 (67.8%)	51 (73.9%)	0.41
ExPEC	alpha-hemolysin	73 (61.9%)	37 (53.6%)	0.28
ExPEC	*cnf*1	74 (62.7%)	36 (52.2%)	0.17
ExPEC	*vat*	99 (83.9%)	55 (79.7%)	0.55
ExPEC	Salmochelin system	**114 (96.6%)**	56 (81.2%)	**<0.001 ***
ExPEC	S pilus	1 (0.8%)	**48 (69.6%)**	**<0.001 ***
ExPEC	Type 1 pilus	118 (100.0%)	69 (100.0%)	1
ExPEC	P pilus	74 (62.7%)	36 (52.2%)	0.17
EXPEC	*ag*43	115 (97.5%)	62 (89.9%)	0.04
EXPEC	*ibe*ABC	0 (0%)	**69 (100%)**	**<0.001 ***
EXPEC	PAI II_536-like_	69 (58.5%)	33 (47.8%)	0.17
STEC ^1^	Shiga like toxin 2	5 (4.2%)	**12 (17.4%)**	**0.004 ***
STEC	*iha*	7 (5.9%)	0 (0.0%)	0.09
EAEC ^1^	*ast*A	**66 (55.9%)**	2 (2.9%)	**<0.001 ***
EAEC	*aat*A	1 (0.8%)	1 (1.4%)	1
EAEC	*pic*	26 (30.5%)	**46 (66.7%)**	**<0.001 ***
Capsule	K1 capsule cluster	**111 (94.1%)**	49 (71.0%)	**<0.001 ***
	Total	118 (100%)	69 (100%)	-

^1^ S1, Sublineage 1; S2, Sublineage 2; Misc., Miscellaneous; ExPEC, extra-intestinal pathogenic *E. coli*; STEC, shiga-toxin-producing *E. coli*; EAEC, Enteroaggregative *E. coli*. * Significantly different (*p* < 0.01). *p*-values were calculated with Fisher’s exact test. S pilus, Shiga like toxin 2, *pic* and *ibe*ABC virulence genes were more frequent in S2 (written in bold). Conversely, the frequency of Salmochelin system, *ast*A and K1 capsule were more important in S1 (written in bold).

## 3. Discussion

We have demonstrated here that the population of *E. coli* ST141 is organized in two main sublineages S1 and S2, one predominantly associated with urinary tract infection (S1) and the other, more frequently associated with intestinal infections (S2). This confirms results generated from a smaller collection [4]. Indeed, for 41 isolates of our collection and previously studied by Gati et al., we noted a good correlation between the L1/L2 Gati’s lineages and our S1/S2 distribution, as all isolates belonging to L2 clustered in the S1 sublineage and possessed the *fim*H5 subtype (Figure 1). Conversely, all isolates belonging to L1 clustered in the S2 sublineage, with different *fim*H subtypes (seven of subtype *fim*H14; three of subtype *fim*H350, two of subtype *fim*H76 and one of subtype *fim*H674) [4] (Appendix A, Figure 1).

The ST141 lineage was highly conserved as all the isolates belonged to B2-phylogroup and had O2:H6 serotype with a restricted number of *fim*H subtypes. This contrasts with other B2 ExPEC successful lineages such as ST95, ST117 and ST131, which display a diversity of O-serogroups [2]. Although, *E. coli* ST141 isolates show various combinations of numerous VFs, as previously observed by Flament et al., we found no clear genomic signature that would indicate an ecological adaptation to a host species (i.e., humans and animals), infections sites, and countries of origin (Appendix A, Figure 1) [9]. However, such diversity of VFs, consistent with the hypothesis that ST141 acts as one of the melting pots within the *E. coli* population, could be evidence of numerous recombinations and the presence of different PAI (pathogenicity islands), which could be interesting to explore in more details [4].

Previously, ST141 had been described as a STEC/UPEC hybrid or an EAEC/UPEC hydrid as some isolates carried *stx*2 or *pic* and *ast*A genes, respectively [4]. However, the majority (13/17) of strains in our ST141 collection (*n* = 187) which genomes contained *stx*2 were those described by Gati et al., and only one of the four additional *stx*2 gene positive strains also possibly had PAI II_536-like_ UPEC virulence factor, questioning the systematic and specifically heteropathogenicity of this clonal group. Moreover, Lindstedt et al. found in a Norwegian collection, a high frequency (64.3%) of *E. coli* strains combining IPEC and ExPEC virulence-associated genes [10]. Another German study revealed that 10.6% of strains isolated from UTIs harbored at least one IPEC virulence factor [11]. Given these conflicting data, the frequency of heteropathogenicity in *E. coli* remains to be clarified, as well as the involvement of different lineages (notably ST141) in this trait.

The fact remains that the ST141 clone has many virulence factors identified in silico. Taking into account the concept of antagonistic pleiotropy and epistatic interactions, a study aiming to characterize the expression of these, in particular in vivo, in an animal model or in vitro, by the capacity of the strains of our collection to produce biofilm could be envisaged in a future work [2].

Nevertheless, *E. coli* ST141 is commonly involved in human diseases and its incidence may reach a significant level. Indeed, ST141 lineage belonged to the 12 most frequent STs involved in bacteremia in France in 2014 [3]. In 2020, ST141 was quoted as one of the most common B2-ExPEC in France [9]. However, incidence of *E. coli* ST141 may vary between countries since a higher incidence of ST141-associated infections was found in France, compared to Spain [9]. In addition, *E. coli* ST141 has been found as the most frequent *E. coli* lineage responsible for ventilator-associated pneumonia over the 2012–2014 study period [12]. Interestingly, in a previous study, Philipps-Houlbracq et al., identified the antigen-43 (Ag43) as significantly associated in pneumonia pathogenesis [13]. It is important to note that Ag43 is widely represented among our collection (like most of B2 *E. coli*) as 97.5% (115/118) and 89.9% (62/69) of strains from sublineage 1 and 2, respectively, harbored this gene (Table 2).

Similarly, the panel of antibiotic resistance genes in the lineage ST141 was large with many combinations (Appendix A, Figure 3). However, we could not identify a multi-resistant epidemic sub-lineage, such as that was observed with ST131 (i.e., C2-H30 producing CTX-M-15) [5]. Indeed, our study depicted a situation where *E. coli* ST141 might only contribute to limited locally spread of ESBL-encoding genes in the community, since the majority of ESBL strains isolated in our hospital form a cluster of strain separated by less than 30 SNPs, suggesting a local limited dissemination of an ESBL clone, although these strains have been isolated from patients with non-obvious epidemiological links (Figure 2) [7].

Likewise, although *E. coli* ST141 was capable of acquiring resistance genes to last resort antibiotics such as carbapenems (*bla*_OXA-48_, *bla*_VIM-1_) and colistin (*mcr*-1), we did not find evidence of the spread of such MDR clones.

## 4. Materials and Methods

### 4.1. ST141 Genome Collection

The genomes of 187 isolates representing ST141 *E. coli* group were obtained from various sources collected over a 30-year period (Appendix A). 

Firstly, 13 non-duplicate ESBL-producing ST141 *E. coli* isolates, collected in our University hospital, in inpatients, during a previous prospective observational cohort study, between 02/2015 and 01/2017 have been paired-end sequenced with Illumina NextSeq at 2 × 150 bp and included [7]. Then, four supplemental non-ESBL-producing ST141 *E. coli* responsible for bloodstream infection, isolated in 2014 in our hospital, have also been fully sequenced and added [3]. Finally, we collected all the ST 141 *E. coli* genomes available in public databases in March 2020 (NCBI and ENA). 

### 4.2. Genome Analysis

Raw reads were first trimmed using Sickle and the 187 genomes were de novo assembled from reads using SPADES, as previously described [14,15]. From these assemblies, we determined in silico: MultiLocus Sequence Typing (MLST) according to the Achtman scheme, O:H serotype, *fim*H type, and phylogroup [8,16]. In the same way, we searched and identified antibiotic resistance genes with ResFinder database, and putative virulence factors (VFs) genes using the VFDB database, which compiles most *E. coli* VFs related to adhesion/invasion, autotransporter system, *fim*bria or flagella expression, iron uptake, serum resistance and toxicity [17,18]. A SNP (Single Nucleotide Polymorphism) call variant was performed against a fully sequenced ST141 *E. coli* genome (NCBI biosample accession number SAMN10740161) used as reference genome using BACTSNP [19]. After recombination curation with Gubbins, a maximum likelihood phylogenetic tree was then inferred from the resulting SNP-based pseudogenomes using RaxML [20,21]. A Cluster Picker analysis was processed to identify phylogenetic clusters (Appendix A) [22]. The tree and corresponding metadata information were visualized with iTOL [23].

### 4.3. Statistical Analysis

To evaluate the distribution of isolates according to their origin and the distribution of VF genes, variables were examined by univariate analysis using the Fisher’s exact test. All statistical tests were two tailed, and *p*-value < 0.01 was considered statistically significant.

## 5. Conclusions

Using a genome-based methodology, we depicted in our study the population structure of *E. coli* ST141 group using a large collection of 187 genomes. Our findings did not confirm the initial hypothesis of an emerging ESBL-producing ST141 sublineage, but rather demonstrated the spread of a subgroup of isolates, showing a closer relatedness, on a regional scale. Nonetheless, we found that *E. coli* ST141 readily acquires and cumulates VF encoding genes and that its prevalence had recently increased either in both extra-intestinal and intestinal diseases. Considering this, there is a need to monitor the spread of this clonal group that has the potential to largely spread in the community and whose involvement in human disease seemed to increase.

## Figures and Tables

**Figure 1 antibiotics-12-00382-f001:**
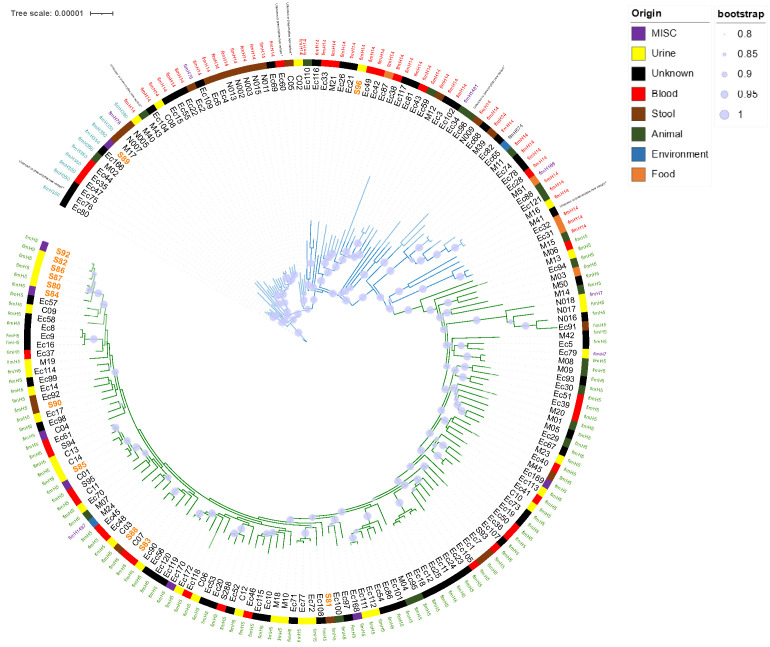
Representation of the two main *E. coli* ST141 sublineages obtained from maximum likelihood phylogeny SNP-based free-recombinant tree inferred from alignment to a fully sequenced reference strain of *E. coli* ST141.

**Figure 2 antibiotics-12-00382-f002:**
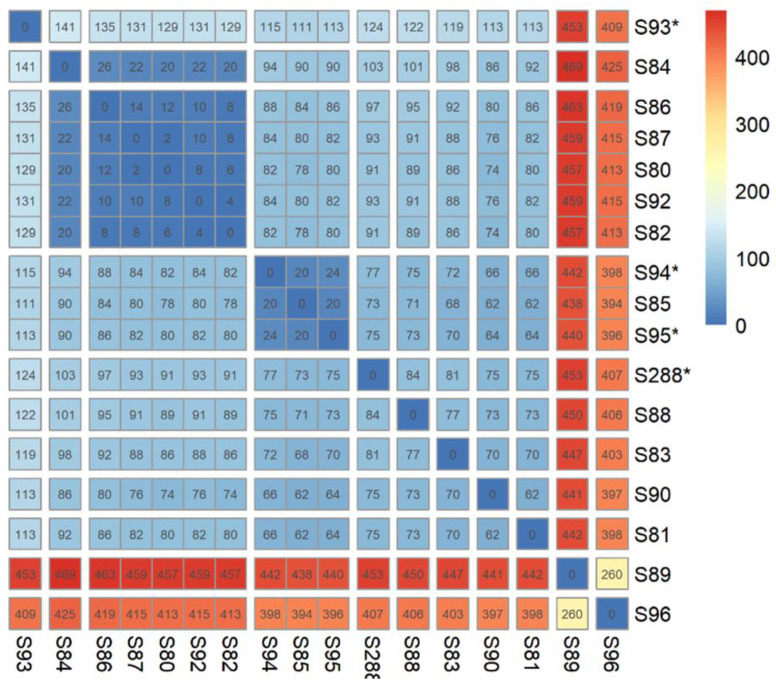
Distance matrix (SNP) of strains from our local collection (*n* = 17). Isolates identified with «*» symbol had no bla_ESBL_ gene. Five of our isolates were less than 15 SNP (S86, S87, S80, S92 and S82) or 26 (S84) from the nearest isolate.

**Table 1 antibiotics-12-00382-t001:** Origin and sublineage distribution of *E. coli* ST141 isolates (*n* = 128).

Source	Data Source	S1 ^1^	S2 ^1^	*p*-Value	Total
Human	Urine	31 (38%)	5 (11%)	0.0009 *	36 (28%)
Human	Blood	22 (27%)	11 (23%)	0.6806	33 (26%)
Human	Diarrhea	3 (4%)	10 (21%)	0.0041 *	13 (10%)
Human	Stool	5 (6%)	5 (11%)	0.4962	10 (8%)
Human	Misc. ^1^	7 (9%)	1 (2%)	0.2564	8 (6%)
Non-human	Animal	11 (14%)	10 (21%)	0.3231	21 (16%)
Non-human	Food	1 (1%)	4 (9%)	0.0604	5 (4%)
Non-human	Environment	1 (1%)	1 (2%)	1.000	2 (2%)
	Total	81 (100%)	47 (100%)	-	128 (100%)

^1^ S1, Sublineage 1; S2, Sublineage 2; Misc., Miscellaneous. * Significantly different (*p* < 0.01). *p*-values were calculated with Fisher’s exact test. Strains with missing data (source and/or data source, *n* = 59/187) were omitted and human and non-human isolates differentiated.

## Data Availability

All ST141 genomes from strains recovered from Besançon (local collection, *n* = 17) are publicy available through the NCBI BioProject PRJNA667655.

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
