# Peer review of "Genotypic Characteristics and Antimicrobial Resistance of Escherichia coli ST141 Clonal Group"

_antibiotics, 2023, doi:10.3390/antibiotics12020382_

Round 1

Reviewer 1 Report

Emery et al. give interesting molecular-biological insights in Escherichia coli ST141 regarding presence and distribution of antimicrobial resistance genes and virulence factors. ST141 belongs to the ExPEC lineage and is frequent but less than ST139. They analysed genomic sequences of 187 isolates at which 170 were from public databases and 17 ST141 strains were isolated during 2015 and 2017 at the University Hospital. Out of these 17 isolates 13 were ESBL producers. The authors followed the question to identify a relevant new emerging sublineage. The answer is that there is no emerging sublineage compared with the lineage ST131, but they showed a spread of a subgroup with regional scale. There are two sublineages, one associated with urinary tract infection and the scones with intestinal infections. ST141 was highly conserved, but contained many virulence factors. There is no indication of a prevalence to specific virulence or toxin genes, and there is no correlation to the ecological adaption, infections and origins. Although no emerging sublineage, the conclusion of the results is the need to monitor new emerging clones in order to avoid great spreads of bacterial subtypes in distinct regions with the potential to cause infections and diseases.

The manuscript is well written and the data analysed are easy to follow. The data analysis included a sufficient number of gene sequences to be able to make a definite statement.      

Figure 1 is blurred and should be presented in a better resolution.

And here are some detail remarks:

  • Line 102: The letter “D” in the beginning of the word is bold.
  • Line 131: 11.2 % (no comma)
  • Line 136: number 34/41. The percentage is missing.

Author Response

Response to Reviewer 1 Comments

Figure 1 is blurred and should be presented in a better resolution.

A higher resolution version of the figures has been uploaded in the corrected manuscript.

And here are some detail remarks:

  • Line 102: The letter “D” in the beginning of the word is bold.
  • Line 131: 11.2 % (no comma)
  • Line 136: number 34/41. The percentage is missing.

Done.

Thanks you for the review

Reviewer 2 Report

This study aims to characterize the ST141 E. coli population from different origins, through whole-genome sequencing, and in Silico comparative analysis

The data reported in the manuscript are scientifically interesting and add information to the state of knowledge about extraintestinal pathogenic E. coli (ST141), especially regarding its phylogenetic, virulence, and antimicrobial resistance characteristics. 

Nevertheless, some issues should be addressed before.

1- The title does not properly disclose the aim of the study 

2- The introduction should be more detailed, namely the pathotype classification. Some data should be added regarding the most critical pathogenic STs.

3- In the conclusion, the authors stated “Our findings did not confirm the initial hypothesis of an emerging ESBL-producing ST141 sublineage”, but it’s not clear to me where they have introduced this hypothesis (or it’s not clearly written).

4- Please provide the correlation between phenotypic and genotypic antibacterial profiles (at least for your strains). 

5- How do authors explain the discrepancies between their results and those of Gati, since both studies used the same available WG sequences?

6- Figures need to be improved: Figure 1 is too small to read and has low resolution. Figure 3 is too small to read.

7- Please indicate and explain under the Table S1 (correspondence vs Gati’s study file) the used red color and the asterisk

8- Line 33: Please italicize “E. coli

9- Please italicize the scientific name of all species mentioned in the reference list

Author Response

Response to Reviewer 2 Comments

1- The title does not properly disclose the aim of the study 

Done. We propose the following title : « Genotypic Characteristics and Antimicrobial Resistance of Escherichia coli ST141 clonal group»

2- The introduction should be more detailed, namely the pathotype classification. Some data should be added regarding the most critical pathogenic STs.

Done. The changes were made directly to the manuscript, line 31 to 55.

3- In the conclusion, the authors stated “Our findings did not confirm the initial hypothesis of an emerging ESBL-producing ST141 sublineage”, but it’s not clear to me where they have introduced this hypothesis (or it’s not clearly written).

Done. The changes were made directly to the manuscript, line 69 to 71. By the way, the title has been changed to reflect this and give greater emphasis to the hypothesis of the emergence of a MDR clone.

4- Please provide the correlation between phenotypic and genotypic antibacterial profiles (at least for your strains). 

Done. The clarification was suggested directly in the manuscript, line 223 to 226.

5- How do authors explain the discrepancies between their results and those of Gati, since both studies used the same available WG sequences?

The differences observed may be related to the difference in size between the 2 studies. Indeed, we included 187 genomes of which only 41 had been studied by Gati. In our collection, we have an “over-representation” of the S1 sub-lineage (n = 118) compared to S2 (n = 69). In Gati’s study, the distribution of strains between the 2 lineages was more balanced (S1 = 19 strains; S2 = 22 strains). By the way, we have corrected a typo (line 286).

6- Figures need to be improved: Figure 1 is too small to read and has low resolution. Figure 3 is too small to read.

A higher resolution version of the figures has been uploaded in the corrected manuscript.

7- Please indicate and explain under the Table S1 (correspondence vs Gati’s study file) the used red color and the asterisk

It is a formatting oversight. There is no particular significance. This has been corrected.

8- Line 33: Please italicize “E. coli

Done

Thanks you for the review

Reviewer 3 Report

The article "E.Coli ST141: genomic characterization and population structure" provided the precise point about the distribution and diversity of the population and several other lineages through its phylogeny. This paper is interesting to read.

There are a few mistakes to be corrected.

1. Line 18 - SNP abbreviation

2. Line 22-25 - Make it easy to understand for readers.

3. Line 45 - Change it to patients

4. Line 60 - It is necessary to mention the strain as most ancient? The word "most" was mentioned too many times. Please avoid it if possible.

5. Line 70 - MLST abbreviation 

6. Figure 1: Increase the resolution. Maintain the dpi mentioned in the author's guidelines.

7. Line 102 and 103: Add the line next to the table title and 'Distance' undo the bold.

8. Line 131: you isolated the gene from the environment? or is it compared with the reference? it is hard to understand.

9. Figure 3: Add line 151 to the title

10. Table 2: Alignment should be corrected.

Author Response

Response to Reviewer 3 Comments

  1. Line 18 - SNP abbreviation

Done

  1. Line 22-25 - Make it easy to understand for readers.

Done. The changes were made directly to the manuscript, line 24 to 26.

  1. Line 45 - Change it to patients

Done

  1. Line 60 - It is necessary to mention the strain as most ancient? The word "most" was mentioned too many times. Please avoid it if possible.

Done. The changes were made directly to the manuscript, line 80 to 82.

  1. Line 70 - MLST abbreviation 

Done

  1. Figure 1: Increase the resolution. Maintain the dpi mentioned in the author's guidelines.

A higher resolution version of the figures has been uploaded in the corrected manuscript.

  1. Line 102 and 103: Add the line next to the table title and 'Distance' undo the bold.

Done. Hope this is ok.

  1. Line 131: you isolated the gene from the environment? or is it compared with the reference? it is hard to understand.

Done. The changes were made directly to the manuscript, line 198 to 199.

  1. Figure 3: Add line 151 to the title

Done. Hope this is ok.

  1. Table 2: Alignment should be corrected.

Done. Hope this is ok.

Thanks you for the review

Reviewer 4 Report

The authors could describe:

1. significance and meaning of fimH based subtyping

2. Figure 1 - poor resolution figures not publication quality

3. Figure 3 - increased resolution of image.

4. phenotypic assays such as biofilm formation, motility or MIC to support observed genotypes of atleast few of ST141 isolates 

Author Response

Response to Reviewer 4 Comments

  1. Significance and meaning of fimH based subtyping

Done. The changes were made directly to the manuscript, line 90 to 99.

  1. Figure 1 - poor resolution figures not publication quality

A higher resolution version of the figures has been uploaded in the corrected manuscript.

  1. Figure 3 - increased resolution of image.

A higher resolution version of the figures has been uploaded in the corrected manuscript.

  1. Phenotypic assays such as biofilm formation, motility or MIC to support observed genotypes of at least few of ST141 isolates

The subject is under consideration for future work. It is now suggested in the manuscript line 308 to 313 .

Thanks you for the review